# Sentinel-2 Exposed Soil Composite for Soil Organic Carbon Prediction

**Klara Dvorakova** [1,*], **Uta Heiden** [2] **and Bas van Wesemael** [1]

1   Georges Lemaître Centre for Earth and Climate Research, Earth and Life Institute, Université Catholique de Louvain, 1348 Louvain-la-Neuve, Belgium; bas.vanwesemael@uclouvain.be
2   German Aerospace Center (DLR), Remote Sensing Technology Institute (IMF), Oberpfaffenhofen, 82234 Wessling, Germany; uta.heiden@dlr.de
*   Correspondence: klara.dvorakova@uclouvain.be

**Abstract:** Pilot studies have demonstrated the potential of remote sensing for soil organic carbon (SOC) mapping in exposed croplands. However, the use of remote sensing for SOC prediction is often hindered by disturbing factors at the soil surface, such as photosynthetic active and non-photosynthetic active vegetation, variation in soil moisture or surface roughness. With the increasing amount of freely available satellite data, recent studies have focused on stabilizing the soil reflectance by building image composites. These composites tend to minimize the disturbing effects by applying sets of criteria. Here, we aim to develop a robust method that allows selecting Sentinel-2 (S-2) pixels with minimal influence of the following disturbing factors: crop residues, surface roughness and soil moisture. We selected all S-2 cloud-free images covering the Belgian Loam Belt from January 2019 to December 2020 (in total 36 images). We then built nine exposed soil composites based on four sets of criteria: (1) lowest Normalized Burn Ratio (NBR2), (2) Normalized Difference Vegetation Index (NDVI) < 0.25, (3–5) NDVI < 0.25 and NBR2 < threshold, (6) the 'greening-up' period of a crop and (7–9) the 'greening-up' period of a crop and NBR2 < threshold. The 'greening-up' period was selected based on the NDVI timeline, where 'greening-up' is considered as the last date of acquisition where the soil is exposed (NDVI < 0.25) before the crop develops (NDVI > 0.25). We then built a partial least square regression (PLSR) model with 10-fold cross-validation to estimate the SOC content based on 137 georeferenced calibration samples on the nine composites. We obtained non-satisfactory results ($R^2$ < 0.30, RMSE > 2.50 g C kg$^{-1}$, and RPD < 1.4, n > 68) for all composites except for the composite in the 'greening-up' stage with a NBR2 < 0.07 ($R^2$ = 0.54 ± 0.12, RPD = 1.68 ± 0.45 and RMSE = 2.09 ± 0.39 g C kg$^{-1}$, n = 49). Hence, the 'greening-up' method combined with a strict NBR2 threshold allows selecting the purest exposed soil pixels suitable for SOC prediction. The limit of this method might be its coverage of the total cropland area, which in a two-year period reached 62%, compared to 95% coverage if only the NDVI threshold is applied.

**Keywords:** soil organic carbon mapping; multispectral data; Sentinel-2; exposed soil composite; greening-up; Normalized Burn Ratio 2

## 1. Introduction

Soil organic carbon (SOC) is crucial for soil functioning, as it affects water and nutrient holding capacity, drainage, aeration, slows down erosion processes, and constitutes the major terrestrial carbon pool. Therefore, SOC was selected as one of the three indicators of the proportion of land that is degraded over total land area in the Sustainable Development Goal (SDG) 15.3.1 [1]. Hence, there is a strong demand for SOC mapping and monitoring, both from environmental and economic perspective. The high spatiotemporal resolution of such information is crucial and has not yet been met by existing soil mapping products such as the harmonized World Soil Database (1:5,000,000, [2]) or the Walloon (Belgium) soil monitoring network CARBOSOL (90 m resolution). Generally, the scale is too coarse and the temporal resolution is more than ten years. After all, changes in SOC are often related to

agricultural management and/or land-use decisions that are taken at the field/farm-scale with a realistic return on investment in mind. Pilot studies have demonstrated the potential of remote sensing for SOC mapping in exposed croplands [3–11]. Vaudour et al. [10], Castaldi et al. [4] and Gholizadeh et al. [11] have used the spectra of the multispectral instrument (MSI) aboard the Sentinel-2 (S-2) constellation to predict SOC contents in croplands of the temperate region. The S-2 constellation is composed of twin satellites S-2A and S-2B in the same orbit but phased at 180° [12], together providing time series with high revisit frequency (five days at the equator). The S-2 MSI has 13 spectral bands covering the visible (Vis)–near infra-red (NIR)–shortwave infra-red (SWIR) spectral range (0.4–2.5 μm). SOC shows a relationship with electromagnetic radiation in all these spectral regions [13]. However, the SOC prediction models established in the above-mentioned studies are all hindered by the atmospheric disturbance (varying with season, clouds, sun azimuth and elevation), as well as by the varying conditions of the surface of the croplands during overflight due to roughness, moisture, or crop residue cover. In fact, the spectrum for an important fraction of the cropland does not reflect the pure soil signal. Moreover, because of crop rotation, which is common practice in Western Europe, the area fraction of exposed soil varies with the acquisition date. The combination of the soil exposure with the soil surface conditions limits the area that is likely to be correctly predicted at any given moment.

Several authors suggested increasing the predicted area by stacking several multi-temporal images [14–19]. Such a multi-temporal mosaic image, or composite image, (i) allows building a more continuous map of exposed soils as it increases the amount of observed exposed soil and (ii) stabilizes the reflectance spectra of the soil. The common approach relies on the empirical definition of a spectral index threshold that is then used to discriminate between soils in suitable and unsuitable conditions. Among these indices are the normalized difference vegetation index (NDVI) [15,18], bare soil index (BSI) [14], normalized burn ratio (NBR2) [16,17] and Sentinel-1 (S-1) derived volumetric soil moisture per pixel (S2WI) [19].

Vaudour et al. [19] used various products for temporal mosaicking of S-2 images to predict SOC in croplands in northern France. They used an S-1 derived volumetric soil moisture separately or in combination with NDVI, NBR2, BSI and S2WI. Overall, the best trade-off between predicted area and model performance was obtained when applying the S-1 derived index to eliminate moist soils. They were able to map 40% of the cropland surface from 13 S-2 spring images acquired over two years for an area characterized by a four-year crop rotation with a good result ($R^2$ ~ 0.5, RPD ~ 1.4, RMSE ~ 3.7 g·kg$^{-1}$). Their study suggests that a number of exposed soil mosaics based on several indicators (moisture, bare soil, roughness, etc.), preferably in combination, might maintain acceptable accuracies for SOC prediction whilst covering a larger area than single-date images [19]. However, they used an S-1 derived moisture index, which is not readily available, and its computation is complex. Moreover, the calculation of this index requires a priori information on the soil moisture condition [20]. Additionally, timelines of SOC prediction models of Vaudour et al. [21] have shown an overall seasonal trend where model performance is positively correlated with solar elevation, thus suggesting that spring images might be favored compared to autumn and winter images in the Northern Hemisphere. Hence, selecting an appropriate image acquisition date might be more important for increasing model performance than the application of an index to discriminate pixels with disturbing effects.

Here, we propose to use a number of indices, which are easy to compute, and which can be obtained from a single satellite, i.e., the S-2 MSI (NDVI and NBR2). The NBR2 index (derived from bands at ~1600 nm and ~2200) has been so far mainly used as indicator for dry crop residues firstly for Landsat8 by Demattê et al. [16], and later for S-2 (B11 and B12) by Castaldi et al. [4]. However, as B11 and B12 of the S-2 MSI cover a broad SWIR range, they are not only strongly correlated to crop residues, but also soil moisture [22]. Daughtry and Hunt [23] have shown that the absorption feature near 2100 nm related to crop residues is significantly attenuated by water content. This limits the sensitivity of the

NBR2 to detect crop residues on moist soils [24]. We therefore chose to combine NDVI and NBR2 thresholds with an automatic selection of appropriate image acquisition dates based on the crop phenology: the -so-called 'greening-up' method. The methodology is inspired by the green-up and green-down processes described by Liu et al. [25], who used these to define cropping cycles. Here, greening-up is defined as the instant where the crop has been or will shortly be sown but has not yet emerged. We introduce this principle for developing exposed-soil composites, because it is during seedbed conditions that the soil surface is in optimal conditions for spectroscopic analysis of its SOC content: (i) an eventual crust and crop residues have been plowed in and (ii) soils have been harrowed and smoothed. Furthermore, as hardly any crop residues are left on the soil surface at the greening-up stage, the NBR2 index can be used to remove pixels affected by water. We believe that the selection of acquisition date based on development of the cropping calendar provides a more reliable index and is easier to accomplish than empirically defining thresholds for multiple spectral indices used in combination.

## 2. Materials and Methods

### 2.1. Study Site and Sample Collection

The study was conducted in the northern part of Wallonia, Belgium. We focus on the croplands in a 110 km × 33 km rectangle, i.e., an intersection of S-2 tile T31UFS and the croplands extracted from the Land Parcel Information system for Wallonia in 2019 (http://geoportail.wallonie.be, accessed date: 28 January 2021, Figure 1A). The rectangle, which covers a total extent of 3630 km$^2$, comprises 1440 km$^2$ of cropland (Figure 1B) and covers mainly the loam belt region dominated by niveo-eolian deposits. The dominant soils are well-drained, loess-derived haplic Luvisols [26]. The relief is gently undulating with altitudes varying between ~80 m (in the north-west) and ~200 m (in the south-east). The climate is temperate oceanic with mean annual precipitation of 790 mm and with the lowest monthly mean temperature in January (2.3 °C) and the highest monthly mean temperature in July (17.8 °C). Predominant land use is cropland with mainly winter cereals, sugar beet, maize and potatoes grown in a three-year rotation. Most cropland soils are under conventional tillage using a moldboard plow.

A total of 137 surface soil samples (0–10 cm) were randomly collected in October 2018 and September 2019 (Figure 1). These soil samples were collected within the framework of earlier studies, and therefore cover a limited extent of the study area [27]. Each sample consists of five sub-samples collected at random locations within a circle of 1 m radius centered on the geographical position of a sampling plot, which was recorded by a Garmin GPS with 3 m precision. The sub-samples were then thoroughly mixed and stored in a plastic bag. Then the samples were air-dried, gently crushed and passed through a 2 mm sieve. SOC was analyzed by dry combustion, using a VarioMax CN Analyzer (Elementar Analysensysteme GmbH, Hanau, Germany), as detailed in Shi et al. [27]. For samples showing reaction with 10% HCl (5 samples out of the 137), carbonate content was measured using a modified pressure-calcimeter method [28]. Then, SOC was obtained by subtracting the inorganic carbon content from total carbon.

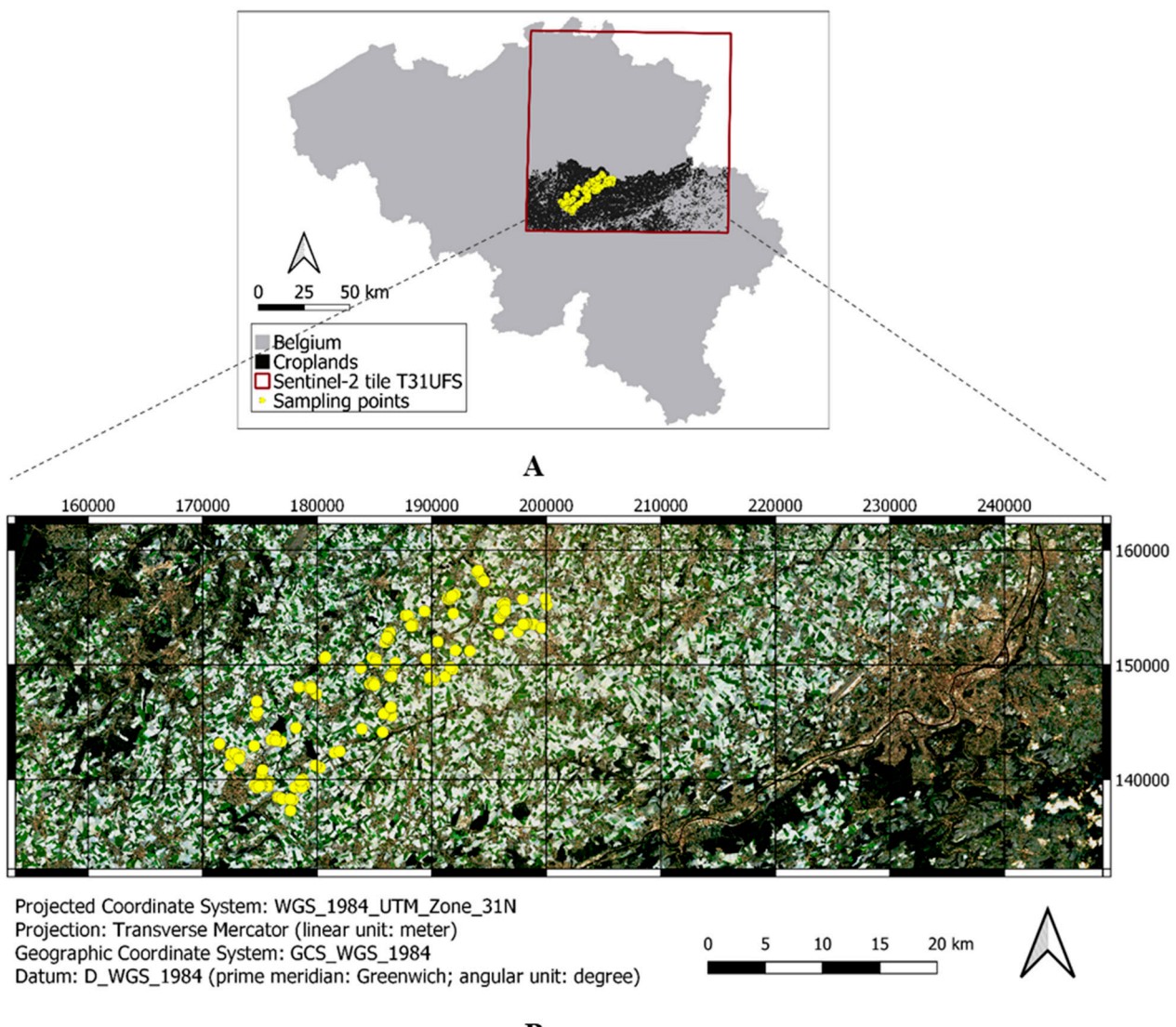

**Figure 1.** (**A**) Location of the Sentinel-2 (S-2) tile T31UFS covering a large part of the Belgian loam with the sample points for calibration and validation (source of the cropland dataset: Service public de Wallonie) and (**B**) Zoom on the study area in the Belgian loam belt and RGB image acquired by the S-2 Multi-spectral Instrument (MSI) on 8 August 2020 (red: 665 nm, green: 560 nm, blue: 490 nm).

### 2.2. Remote Sensing Data

Spectra were obtained using the MSI aboard the S-2A and S-2B platforms, as the S-2 mission is a constellation with twin satellites. The MSI has 13 spectral bands, including four bands of 10 m resolution and six bands of 20 m resolution (Table 1). A cloud-free time series composed of 36 images from 1 January 2019 to 31 December 2020 was obtained from the French land data center (https://theia.cnes.fr, Accessed date: 6 January 2021). For each date, ten bands (B2, B3, B4, B5, B6, B7, B8, B8A, B11 and B12, see Table 1) with correction of slope effects were provided as Level-2A product, i.e., geometrically, radiometrically and atmospherically corrected using the MACCS-ATCOR Joint Algorithm (MAJA) processor. Since the MSI has bands with different spatial resolutions, the images were spatially resampled (nearest neighbor resampling) at 10 m to maximize the level of detail of the S-2 data. The images were then masked using the Walloon cropland map (http://geoportail. wallonie.be/catalogue/81bdf8bc-5968-4fd3-84ca-6be011cddd6.html, accessed date: 28 January 2021).

**Table 1.** Specifications of the Multispectral Instrument (MSI) on board of the Sentinel-2 constellation. Vis = visible, R-edge = red edge, NIR = near-infrared, SWIR = shortwave infrared. In bold are the bands provided by Theia (theia.cnes.fr).

| Spectral Band | Spectral Domain | Central Wavelength (nm) | | Bandwidth (nm) | | Spatial Resolution (m) |
|---|---|---|---|---|---|---|
| | | S-2A | S-2B | S-2A | S-2B | |
| B1 | Vis | 442.7 | 552.2 | 66 | 66 | 60 |
| **B2** | **Vis** | **492.4** | **492.1** | **36** | **36** | **10** |
| **B3** | **Vis** | **559.8** | **559.0** | **31** | **31** | **10** |
| **B4** | **Vis** | **664.6** | **664.9** | **106** | **106** | **10** |
| **B5** | **R-edge** | **704.1** | **703.8** | **15** | **16** | **20** |
| **B6** | **R-edge** | **740.5** | **739.1** | **15** | **15** | **20** |
| **B7** | **R-edge** | **782.8** | **779.7** | **20** | **20** | **20** |
| **B8** | **NIR** | **832.8** | **832.9** | **21** | **22** | **10** |
| **B8A** | **NIR** | **864.7** | **864.0** | **91** | **94** | **20** |
| B9 | NIR | 945.1 | 943.2 | 175 | 185 | 60 |
| B10 | SWIR | 1373.5 | 1376.9 | 21 | 21 | 60 |
| **B11** | **SWIR** | **1613.7** | **1610.4** | **20** | **21** | **20** |
| **B12** | **SWIR** | **2202.4** | **2185.7** | **31** | **30** | **20** |

### 2.3. Spectral Indices

A set of spectral indices was calculated for each S-2 acquisition date: NDVI [29] (Equation (1)) and NBR2 [30] (Equation (2)),

$$\text{NDVI} = \frac{\rho_{\text{NIR}} - \rho_{\text{Red}}}{\rho_{\text{NIR}} + \rho_{\text{Red}}} \tag{1}$$

$$\text{NBR2} = \frac{\rho_{\text{SWIR1}} - \rho_{\text{SWIR2}}}{\rho_{\text{SWIR1}} + \rho_{\text{SWIR2}}} \tag{2}$$

where $\rho$ is the surface reflectance (%) of the red, near-infrared (NIR) and far shortwave infrared (SWIR) spectral regions (i.e., Red = B4, NIR = B8, SWIR1 = B11 and SWIR2 = B12 for the MSI on board of the S-2 constellation).

Values range between −1 and 1, where higher values of NDVI indicate high green vegetation coverage. Choosing a threshold NDVI value is required for masking green vegetation. The threshold was determined by (i) visually inspecting the S-2 RGB images and by (ii) minimizing the 'salt-and-pepper' patchiness of the resulting mask. Overall, pixels with NDVI values above 0.25 were considered as pixels containing green vegetation. This threshold was kept constant for all 36 S-2 images.

Dvorakova et al. have shown that when soils are dry, NBR2 follows a linear relationship with crop residue cover, however, in the case of moist soils, no correlation with residue cover could be found [24]. We, therefore, assume that NBR2 reacts both to crop residues and soil moisture, where high values of NBR2 indicate soils that are moist and/or are covered by crop residues, but we do not make any assumptions about the form of this relationship. Hence, as setting a threshold for the NBR2 index might be erroneous without relevant field observation, we chose four arbitrary classes of NBR2: (i) no threshold, (ii) below 0.15, (iii) below 0.10 and (iv) below 0.07. Additionally, weekly meteorological data from the Royal Meteorological Institute (RMI; https://www.meteo.be/en/brussels, accessed date: 15 December 2020) weather stations Ernage and Bierset were retrieved from January 2019 until December 2020. These data were used to compare the response of NBR2 to rainfall events.

The NDVI and NBR2 indices are used here to detect soils that are likely to be exposed. Without field observation, however, it is not possible to ensure that these soils are in fact exposed. For the sake of simplicity, the 'exposed soil' terminology is used to describe soils with NDVI and NBR2 indices below the selected thresholds, suggesting that they are likely to be exposed.

### 2.4. Methods for Creating Composites of Exposed Soils

The first step was generating NDVI and NBR2 layers for each of the 36 cloud-free S-2 images using Equations (1) and (2). An NDVI threshold obtained by expert judgment (Section 2.3) was then applied to the NDVI layer, which was converted into 0 and 1. Hence, pixels with NDVI ≤ 0.25 were reclassified to value 0, and pixels with NDVI > 0.25 became 1. The binary NDVI layer and NBR2 layer were then stacked chronologically, in $BinaryNDVI_{stack}$ and $NBR2_{stack}$. Any manipulation that results in the formation of a composite was performed on $BinaryNDVI_{stack}$ and $NBR2_{stack}$, until the final extraction of S-2 MSI spectra for the selected exposed soil pixels. This significantly reduced the processing time. The S-2 reflectance data for pixels that respect the conditions given by the composites were extracted for bands B2, B3, B4, B5, B6, B7, B8, B8A, B11 and B12, forming full S-2 spectra. If multiple dates per pixel were selected, the S-2 reflectance data were averaged by band, in order to obtain one final spectrum.

#### 2.4.1. Spectral Indices-Only Approach

The first approach implies that the pixels to be used in the final composite image are selected on a date-independent basis, solely based on a selected threshold value of NDVI and/or NBR2 indices. Overall, five composites based on spectral indices only were proposed (i.e., A–E in Figure 2):

*Driven by the lowest NBR2 value amongst an S-2 time series (Composite A).*

Firstly, for each pixel, S-2 acquisition dates where $BinaryNDVI_{stack}$ equals 0 were kept. An NBR2 value per date was then extracted per pixel, resulting in a vector of 36 NBR2 values for each pixel of the image. Amongst these values, the date with the lowest NBR2 value was maintained for a given pixel.

*Driven by NDVI threshold value only (Composite B).*

For each pixel, S-2 acquisition dates where $BinaryNDVI_{stack}$ equals 0 were kept.

*Driven by NDVI and NBR2 threshold values (Composite C, D and E).*

Firstly, for each pixel, S-2 acquisition dates where $BinaryNDVI_{stack}$ equals 0 were kept. Then, for a given pixel, NBR2 information was extracted for those selected dates from $NBR2_{stack}$. In the next step, $NBR2_{0.15}$, $NBR2_{0.10}$ and $NBR2_{0.07}$ thresholds were applied for composites C, D and E respectively, where any S-2 acquisition dates where NBR2 value was above the specified thresholds, were eliminated.

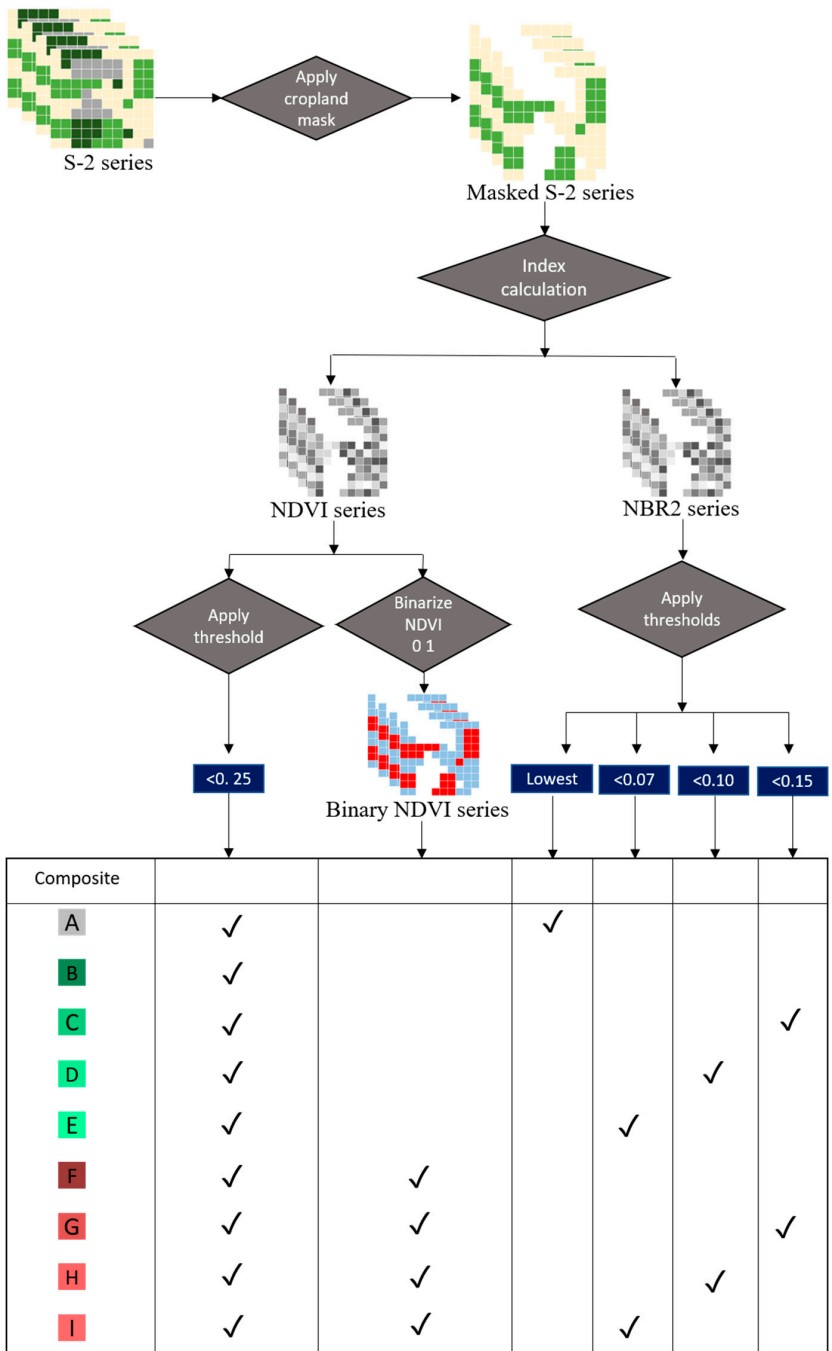

**Figure 2.** General flowchart for the composite-making approach.

## 2.4.2. Greening-Up Approach

The greening-up approach relies on the assessment of the temporal sequence of NDVI binary values. The pixels to be used in the final composite image were selected based on the cropping calendar. We hypothesize that a window exists in the cropping calendar, during which the surface conditions reflect the spectral signature of the soil. This occurs when soils are dry and in seedbed condition, and therefore are harrowed and without any green vegetation or crop residues. The highest likelihood for the occurrence of seedbed conditions was considered to be the last moment in the NDVI sequence, during which vegetation is on the verge of starting to grow (Figure 3). Mathematically, this is translated as the intersection of the NDVI curve with the threshold selected for exposed soils (here NDVI < 0.25) before the NDVI reaches a local maximum (e.g., $t_5$ in Figure 3). We propose

another four composites based on the greening-up approach (combined with an NBR2 index; i.e., F–I in Figure 2).

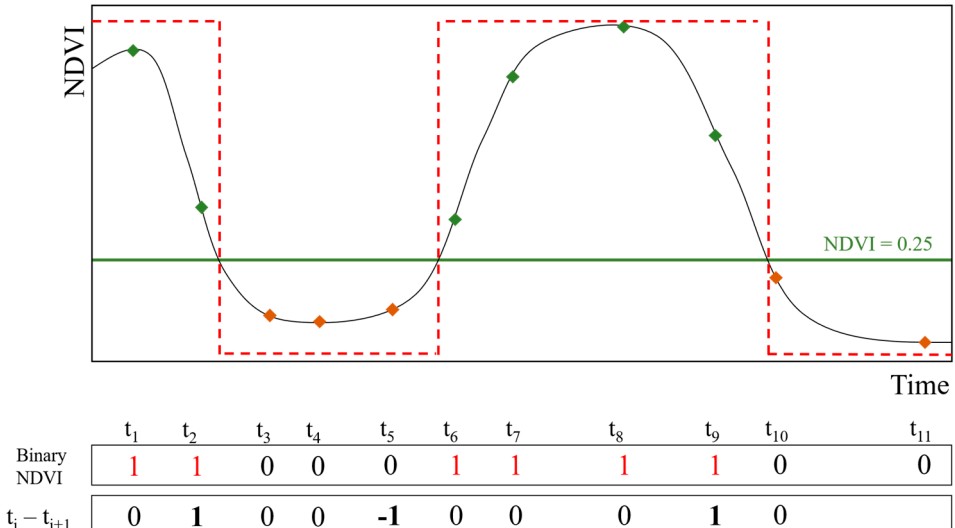

**Figure 3.** Illustration of the recognition of the greening-up moment based on an NDVI time series. The points indicate an imaginary satellite image acquisition, the black curve is the shape of an ideal NDVI evolution of cropland. The red dashed line is binary representation of NDVI: NDVI $\leq$ 0.25 becomes phase 0 (red points), NDVI > 0.25 becomes phase 1 (green points). In this case, point at $t_5$ is considered to be at the greening-up stage.

*Driven by the Greening-up approach (Composite F, in Figure 2).*

For each pixel, the temporal sequence of NDVI binary values was extracted from BinaryNDVI$_{\text{stack}}$ (Equation (3)).

$$\text{BinaryNDVI}_{t_i} - \text{BinaryNDVI}_{t_{(i+1)}} \tag{3}$$

where $t \in [1, T]$ is an image acquisition date, and hence varies from 1 to 36. Each $t_i$ for which Equation (3) equals $-1$ (see Table 2 for combinations) was considered as greening-up. In order to avoid false greening-up stages which are caused by a short-term increase in NDVI which does not result in a full cropping cycle, a lower limit was applied to the length of a cropping cycle. Here, the length of the cropping cycle was set as the duration (in days) which separates the instant $t_i$ when Equation (3) is equal to $-1$ until the instant $t_i$ at which Equation (3) is equal to 1 (in this order). If this time period was shorter than 50 days, we excluded the selected greening-up point. This threshold is modifiable based on the regional phenology. The S-2 spectra for a given pixel were then extracted from the S-2 image acquired at $t_i$ which was detected to be a greening-up moment.

**Table 2.** BinaryNDVI for exposed soil (0) and vegetation (1) and the combinations obtained when applying Equation (3). Exposed soil followed by vegetation (which is considered here as the so-called greening-up moment) has value $-1$.

| BinaryNDVI$_{ti}$ | BinaryNDVI$_{t(i+1)}$ | BinaryNDVI$_{ti}$ $-$ BinaryNDVI$_{t(i+1)}$ |
|---|---|---|
| Exposed soil (0) | Exposed soil (0) | 0 |
| Vegetation (1) | Exposed soil (0) | 1 |
| Vegetation (1) | Vegetation (1) | 0 |
| Exposed soil (0) | Vegetation (1) | $-1$ |

*Driven by the Greening-up approach combined with NBR2 threshold value (Composites G, H and I, in Figure 2).*

The greening-up was applied, in combination with an NBR2 index threshold value. In such a way, we select pixels in seedbed conditions, and with hardly any crop residue cover. The NBR2 index is then applied to eliminate moist soils. For each selected $t_i$, the NBR2 value was extracted from $NBR2_{stack}$ for a given pixel. NBR2 arbitrary thresholds $NBR2_{0.15}$, $NBR2_{0.10}$ and $NBR2_{0.07}$ were applied for composites G, H and I respectively. If the NBR2 value was above the specified thresholds, the greening-up detected at $t_i$ was eliminated.

### 2.5. Composite Surface Cover

It is important to characterize the fraction of the cropland area that is bare soils for each of the composites. We selected 18,000 points on each S-2 composite in order to diminish the processing time. The points were created within the intersection of the S-2 T31UFS tile with the dataset of Walloon croplands so that only croplands are included (Figure 1). A stratified random sampling method was applied to the soil association map (1:250,000, *Service Public de Wallonie*; http://geoportail.wallonie.be, accessed date: 28 January 2021) in order to create a representative set of points. The presence or absence of exposed soil was calculated for each of the 18,000 points on composites A to I (Figure 2).

### 2.6. Spectral Models for SOC Prediction

VNIR-SWIR spectra were extracted from the S-2 images for all 36 acquisition dates at the locations of the 137 soil samples by means of the bilinear interpolation technique (Figure 1). This method assigns the output cell value by taking the weighted average of four closest cell centers.

*SOC Prediction by Date.*

For each S-2 acquisition date, four calibration subsets were created with (i) all samples with NDVI ≤ 0.25, (ii) all samples with NDVI ≤ 0.25 and NBR2 < 0.15, (iii) all samples with NDVI ≤ 0.25 and NBR2 < 0.10 and (iv) all samples with NDVI ≤ 0.25 and NBR2 < 0.07.

*SOC Prediction by Composite.*

Nine subsets of calibration samples were extracted under the constraints applying to each of the composites described above (Section 2.4 and Figure 2).

The partial least square regression (PLSR) model was then chosen to construct SOC prediction models based on the selected set according to the criteria (Figure 2) from the total sample set (n = 137). The PLSR approach uses the full spectrum to establish a linear regression model where the significant spectral information is contained in a few orthogonal factors, called latent variables (LV) [31,32]. Because a limited number of samples were available, a ten-fold-cross-validation procedure was adopted to estimate the prediction capability of the PLSR model for the training set. The PLSR analyses were performed using the 'pls' package developed in R software [33]. To avoid over- or under-fitting, the optimal number of LV was determined as the one producing a model having the minimal Root Mean Square Error (RMSE) of cross-validation, while the maximum number of LV possible was set to five.

The quality of model fit was assessed using the following parameters: coefficient of determination ($R^2$), Root Mean Square Error (RMSE), and Ratio of Performance to Deviation (RPD) of 10-fold-cross-validation (Equations (4)–(6)):

$$R^2 = \frac{\sum_{i=1}^{n}\left(\hat{y}_i - \bar{y}_i\right)^2}{\sum_{i=1}^{n}\left(y_i - \bar{y}_i\right)^2} \tag{4}$$

$$RMSE = \sqrt{\frac{\sum_{i=1}^{n}(\hat{y}_i - y_i)^2}{n}} \tag{5}$$

$$\text{RPD} = \frac{\text{std}}{\text{RMSE}} \tag{6}$$

where $\hat{y}$ = predicted value, $\bar{y}$ = mean observed value, y = observed values, n = number of samples with i = 1, 2, . . . , n, and std the standard deviation of the observed values.

Thresholds for RPD can be found which classify the models into three categories: non reliable when RPD < 1.4, fair when 1.4 < RPD < 2 and excellent when RPD > 2 [34]. Minasny [35], however, considers these thresholds to be arbitrary. We will, therefore, not use the thresholds as model performance indicators, but provide these for comparison with the literature only.

Additionally, each set of calibration samples was subject to bootstrapping to stabilize the prediction model performance. Bootstrapping consists of repeatedly calculating a given statistic from a series of subsamples obtained by randomly resampling with replacement an initial dataset [36]. Hence, for each set of calibration samples, 100 PLSR models were created, and the final model performance corresponds to the mean of the 100 created models.

Several subsets of calibration points were selected for the S-2 composites to exclude unsuitable spectra from the PLSR analysis. To ensure that the predictive accuracies of various PLSR models fit to each S-2 composite are comparable, one must make sure that the training datasets are comparable. Vašát et al. [37] have shown that training sets with larger variance achieve a more accurate prediction in terms of variance explained. Therefore, Levene's test ('car' package in the R Core Team, 2017 [33]) was used to verify the assumption that variances are equal across all training sets with a significance level of $\alpha$ = 0.05. The SOC training sets were further analyzed by descriptive statistics and frequency histograms.

### 3. Results

#### 3.1. PLSR Models for Single S-2 Acquisition Dates

The SOC content was on average 12.3 g kg$^{-1}$ and was rather variable: variation coefficient (CV = 27.3%; n = 128; Table 2). The prediction accuracy of PLSR models was variable (Figure 4): worst model performance was obtained for 5 August 2020 when no NBR2 threshold was applied ($R^2$ = 0.08 $\pm$ 0.3, RMSE = 3.47 $\pm$ 0.23 g C kg$^{-1}$ and RPD = 0.1 $\pm$ 0.0, n = 71), while the best was obtained for 22 March 2019 with NBR2 threshold 0.07 ($R^2$ = 0.78 $\pm$ 0.13, RMSE = 0.45 $\pm$ 0.11 g C·kg$^{-1}$ and RPD = 2.7 $\pm$ 0.9, n = 21). The application of the various NBR2 thresholds did not always improve the model performance for a given acquisition date. In some cases, however, the model performance yielded much better results ($R^2$ increased by a factor of eight for the image from 5 August 2020 when no NBR2 threshold was compared with an NBR2 of 0.07). Note that when less than 20 calibration samples were available, models were not assumed stable and the PLSR results are therefore not shown.

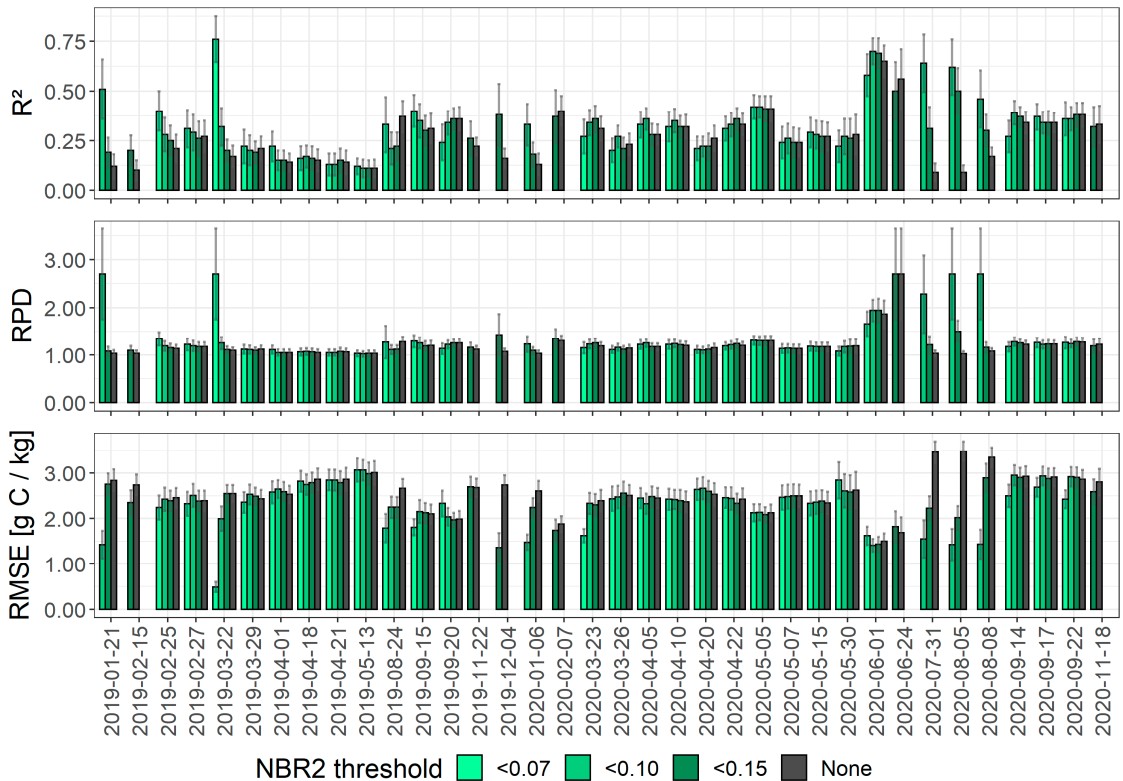

**Figure 4.** Coefficient of determination ($R^2$), Ratio of Performance to Deviation (RPD) and Root Mean Square Error (RMSE) of the Partial Least Squares Regression (PLSR) models applying the Normalized burn ratio (NBR2) thresholds according to Sentinel-2 acquisition date between January 2019 and December 2020. Note that because of a low number of calibration samples (<20), some results are not provided.

### 3.2. PLSR Models for S-2 Composites

Overall, nine S-2 composites were built, which were used to select calibration subsets for the PLSR models (Figure 2). The CV shows a rather stable value for all subsets (min 22.8% for composite H; max 27.3% for composites A, B and C). The Levene's test indicates the homogeneity of variances across the calibration subsets (Table 3). All *p*-values were higher than the significance level of $\alpha = 0.05$, except for composite G against composites A, B and C. These subsets have significantly different variances and should therefore not be compared. The model performance varies with composite type (Figure 5, Table 4). Composite I yields satisfactory results ($R^2 = 0.54 \pm 0.12$, RMSE = $2.09 \pm 0.39$ g kg$^{-1}$ and RPD = $1.68 \pm 0.45$), while the model performance of the other composites is poor ($R^2 < 0.20$).

**Table 3.** *p*-Value of Levene's test of homogeneity of variances between training calibration datasets for the various composites. Null hypothesis: the population variances are equal. In bold: Training set combinations that reject the null hypothesis at $\alpha = 0.05$ and the variances of which are therefore not equal. For interpretation of the composites see Figure 2 and Table 4.

| Composite | A | B | C | D | E | F | G | H | I |
|---|---|---|---|---|---|---|---|---|---|
| A | - | | | | | | | | |
| B | 1.000 | - | | | | | | | |
| C | 1.000 | 1.000 | - | | | | | | |
| D | 0.666 | 0.666 | 0.666 | - | | | | | |
| E | 0.332 | 0.332 | 0.332 | 0.592 | - | | | | |
| F | 0.390 | 0.390 | 0.390 | 0.641 | 0.979 | - | | | |
| G | **0.048** | **0.048** | **0.048** | 0.098 | 0.207 | 0.278 | - | | |
| H | 0.114 | 0.114 | 0.114 | 0.193 | 0.338 | 0.339 | 0.881 | - | |
| I | 0.890 | 0.890 | 0.890 | 0.705 | 0.495 | 0.495 | 0.117 | 0.141 | - |

**Table 4.** Partial Least Squares Regression (PLSR) with 10-fold cross-validation applied to the Sentinel-2 derived composites. In bold is the best model. n = number of calibration samples.

| Composite | Criteria | Descriptive Statistics | | | | | | Tenfold-Cross-Validation | | |
|---|---|---|---|---|---|---|---|---|---|---|
| | | n | Min * | Max * | Mean * | STD * | CV (%) | RMSE * | $R^2$ | RPD |
| A | Lowest NBR2 | 128 | 6.7 | 22.1 | 12.3 | 3.4 | 27.3 | $3.63 \pm 0.36$ | $0.14 \pm 0.03$ | $1.06 \pm 0.06$ |
| B | NDVI < 0.25 | 128 | 6.7 | 22.1 | 12.3 | 3.4 | 27.3 | $3.54 \pm 0.32$ | $0.22 \pm 0.04$ | $1.12 \pm 0.07$ |
| C | NDVI < 0.25 and NBR2 < 0.15 | 127 | 6.7 | 22.1 | 12.3 | 3.4 | 27.3 | $3.46 \pm 0.34$ | $0.22 \pm 0.04$ | $1.12 \pm 0.06$ |
| D | NDVI < 0.25 and NBR2 < 0.10 | 126 | 6.7 | 22.1 | 12.2 | 3.2 | 26.6 | $3.45 \pm 0.30$ | $0.21 \pm 0.04$ | $1.12 \pm 0.06$ |
| E | NDVI < 0.25 and NBR2 < 0.07 | 123 | 6.7 | 21.4 | 12.1 | 3.1 | 25.4 | $3.43 \pm 0.31$ | $0.19 \pm 0.04$ | $1.10 \pm 0.07$ |
| F | Greening-up | 108 | 7.4 | 20.2 | 11.7 | 3.1 | 25.4 | $2.74 \pm 0.23$ | $0.15 \pm 0.04$ | $1.06 \pm 0.08$ |
| G | Greening-up and NBR2 < 0.15 | 91 | 7.4 | 20.2 | 11.6 | 2.7 | 22.9 | $2.43 \pm 0.24$ | $0.14 \pm 0.05$ | $1.06 \pm 0.08$ |
| H | Greening-up and NBR2 < 0.10 | 68 | 7.4 | 20.2 | 11.6 | 2.7 | 22.8 | $2.21 \pm 0.27$ | $0.26 \pm 0.08$ | $1.14 \pm 0.08$ |
| **I** | **Greening-up and NBR2 < 0.07** | **49** | **8.0** | **20.2** | **11.3** | **3.2** | **25.7** | **$2.09 \pm 0.39$** | **$0.54 \pm 0.12$** | **$1.68 \pm 0.45$** |

* expressed in $g\ kg^{-1}$.

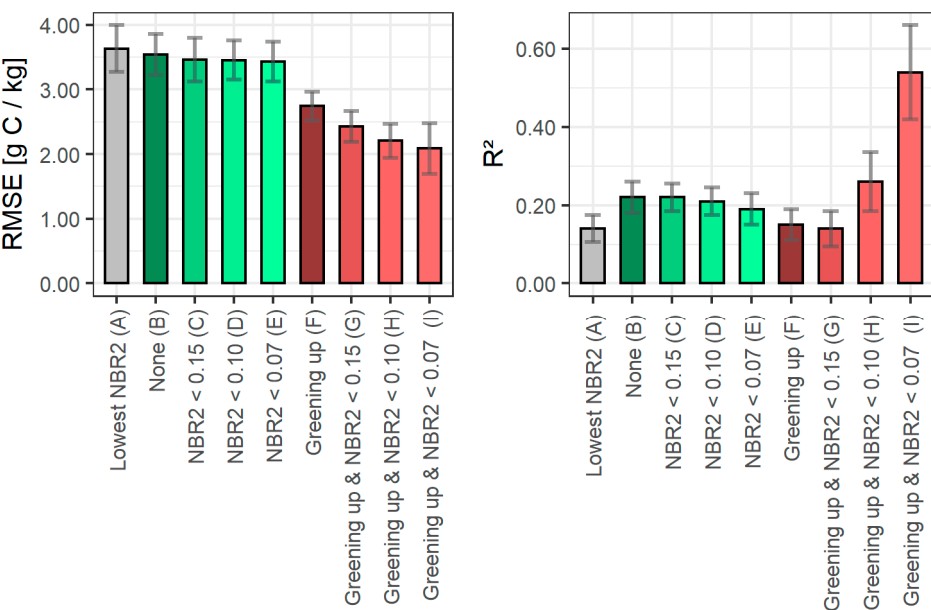

**Figure 5.** Coefficient of determination ($R^2$) and Root Mean Square Error (RMSE) of the Partial Least Squares Regression (PLSR) models applied to Sentinel-2-derived composites based on various criteria.

The number of calibration samples available for the various composites dropped greatly for the combination of the greening-up approach with NBR2 thresholds: for the strictest $NBR2_{0.07}$ threshold (composite I), only 49 samples were available, compared to more than 120 samples for composites A to E (Table 2). Mainly the spring to early autumn acquisition dates provide calibration points for the greening-up composite I (Figure 6).

An example of NDVI and NBR2 time series from January 2019 to November 2020 is provided for two pixels (Figure 7). The main crop type in 2019 is maize (Figure 7A) and winter wheat (Figure 7B). The selection strategy for the dates to be included in composites B, E, F and I is also shown (Figure 7). It highlights the narrowing down of the acquisition date selection. The same point (example of Figure 7B) is included 13 times in composite B, while only once in composite I.

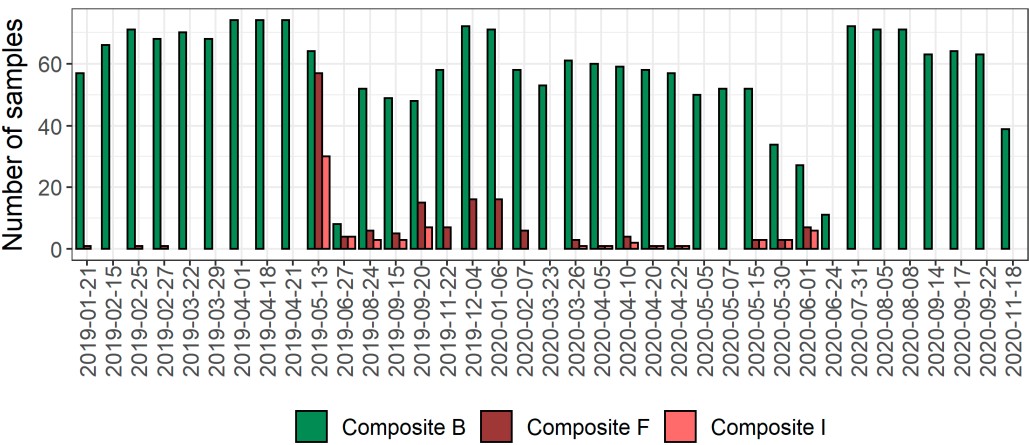

**Figure 6.** Evolution of the number of calibration samples available for each Sentinel-2 cloud-free image between January 2019 and December 2020 based on the composite criteria selection strategy.

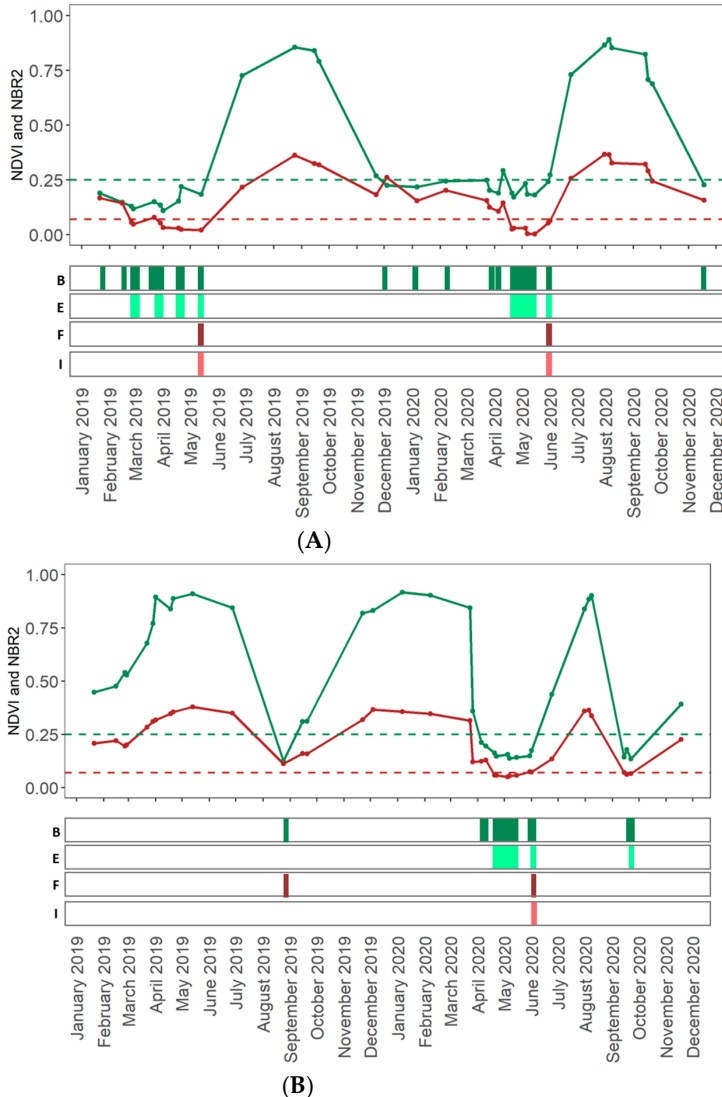

**Figure 7.** Normalized difference vegetation index (NDVI, in green) and Normalized burn ratio (NBR2, in red) series for two pixels: maize field in 2019 (**A**) and winter wheat field in 2019 (**B**). The colored vertical bars represent the inclusion of the point in the composite B, E, F or I (for interpretation of the composites see Table 4 and Figure 2). The dashed horizontal lines are NDVI = 0.25 (in green) and NBR2 = 0.07 (in red).

### 3.3. Surface Area Coverage by the Different Composites

For composite B, which only takes into account the NDVI threshold for bare soils, the soil exposure was above 30% for all acquisition dates except for the middle of the growing season (end of May until July, Figure 8A). When applying both NDVI and NBR2 thresholds (composite E) the winter months disappeared together with an overall reduction of exposed soil pixels. Composite I based on the combination of the greening-up approach and $NBR2_{0.07}$ threshold allowed for a very limited soil exposure. Only five dates in spring and autumn out of the 36 S-2 images produced more than 10% soil exposure (Figure 8A). A link between the weekly rainfall and the decrease in exposed surface when applying the $NBR2_{0.07}$ threshold is observed during winter months (Figure 8B). The exposed surface on composite E drops compared to composite B between October 2019 and February 2020. No particular link is observed between weekly precipitation and exposed soil surface on composite E during the months of April, May and June, probably because the surface of the soil dries up quickly in late spring and early summer. In contrast, the S-2 overflight of three images from months of July and August (24 August 2019, 31 July 2020, 8 August 2020) took place where no weekly precipitations were measured. Yet, the exposed soil surface on these three images dropped considerably after applying the $NBR2_{0.07}$ threshold. This is likely due to the presence of dry residues of cereals on the soil surface. The cumulative percentage of croplands with exposed soils showed that for a two-year period, composite B yielded 95% soil exposure, composite E 87%, composite F 88%, and composite I 62% (Figure 9). For the greening-up approach, the biggest increase occurred in May and September 2019, which corresponds to the seeding periods of summer and winter crops. Finally, for all pixels included in composite I, the crop type in 2019 was extracted from the dataset of Walloon croplands (Figure 10). The results suggest that for the six main crops, winter cereals and maize are underrepresented in composite I, while peas, sugar beet and potatoes are overrepresented.

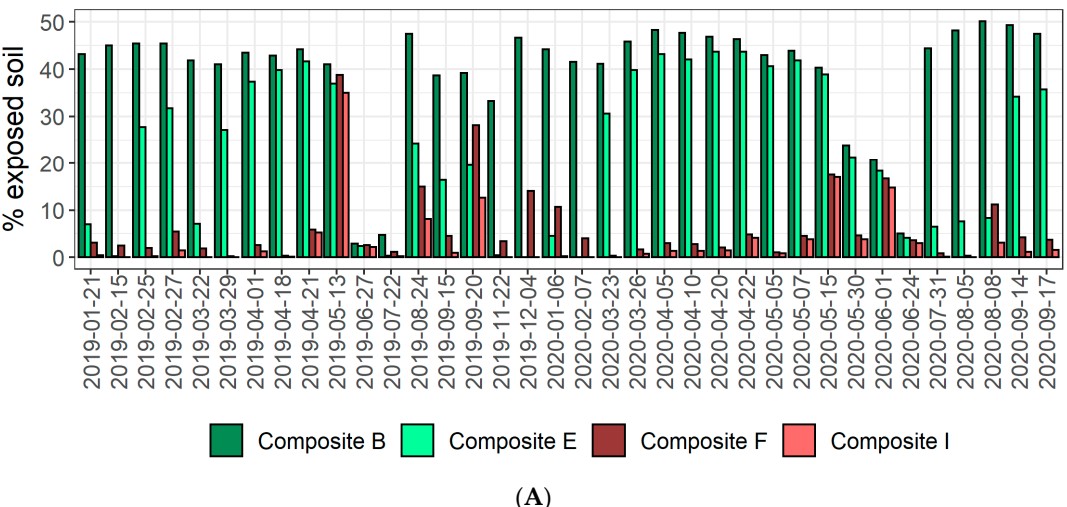

**(A)**

**Figure 8.** *Cont.*

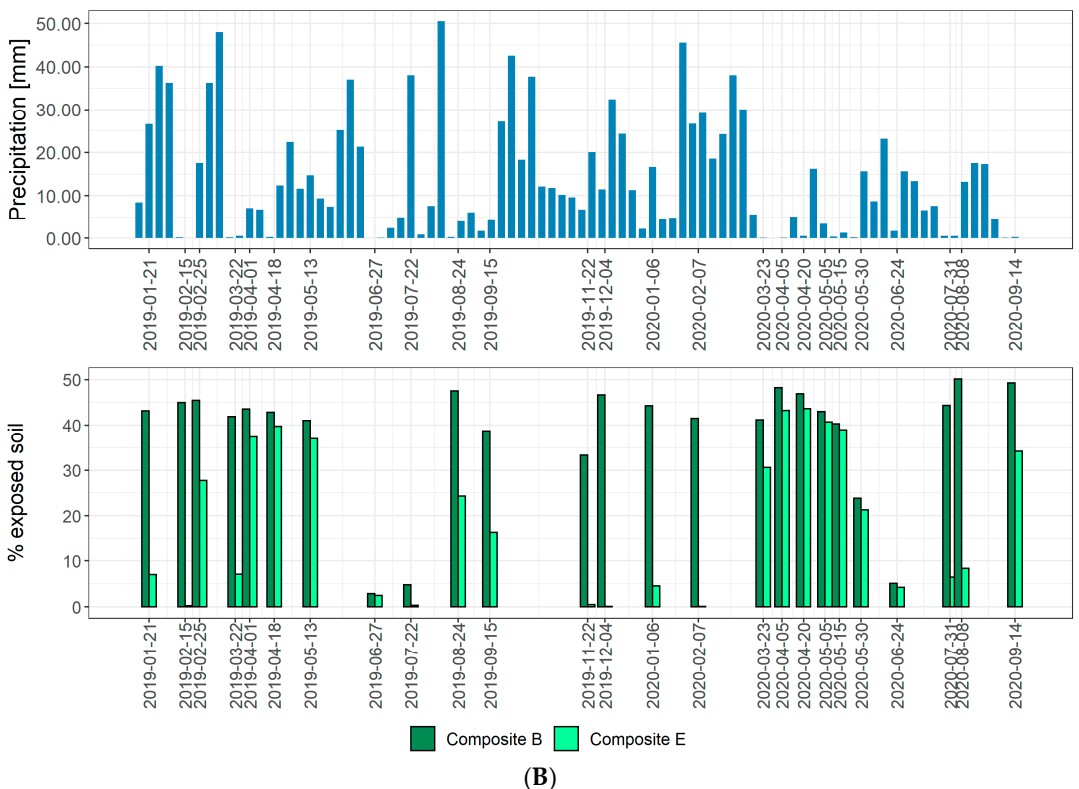

(**B**)

**Figure 8.** Evolution of the percentage of exposed soil on each Sentinel-2 cloud-free image between January 2019 and December 2020 based on the composite criteria selection strategy (**A**) and the timeline of cumulated weekly precipitation within the area and the percentage of exposed soil on composites B (NDVI only) and E (NDVI and NBR2). For visual reasons, some Sentinel-2 acquisition dates were removed from (**B**). Source of the weather data: The Royal Meteorological Institute of Belgium.

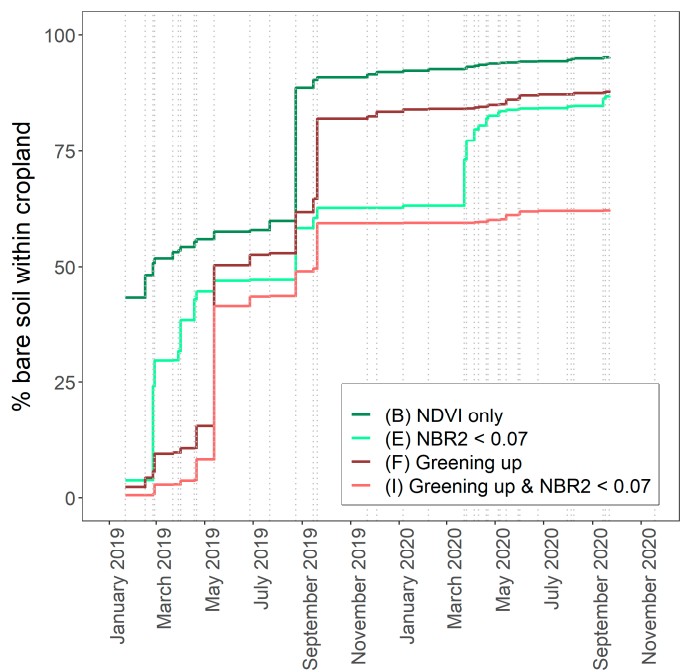

**Figure 9.** Cumulative percentage of exposed croplands extracted from all cloud-free Sentinel-2 (S-2) images between January 2019 and December 2020, depending on the composite selection criteria (B, E, F or I). The vertical dashed lines are the S-2 acquisition dates.

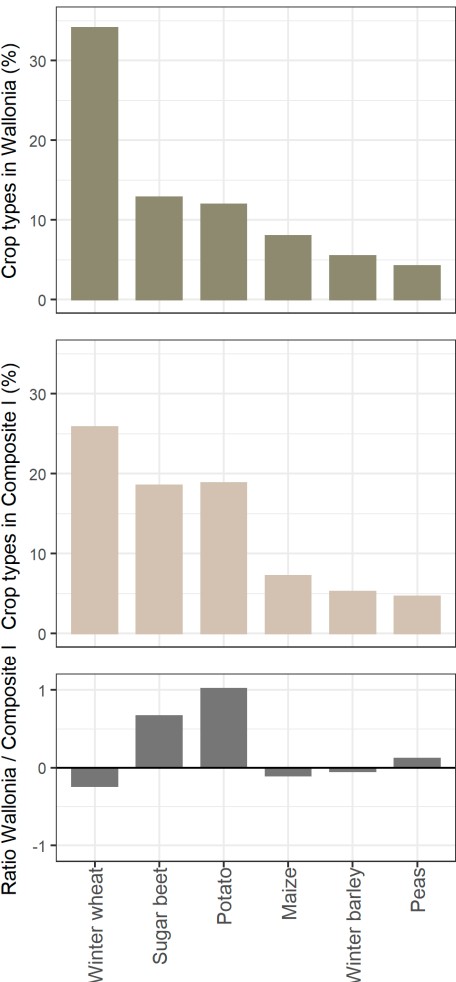

**Figure 10.** The percentage of surface cover of the main six types of crops in the Walloon region (**top**), the representation of the main six types of crops in the Composite I (**middle**) and the logarithm of the ratio between the two (**bottom**).

## 4. Discussion

As several studies have shown [19,21], the SOC prediction model performance from S-2 images relies on the selection of the acquisition date. This selection depends on a number of factors that influence the state of the soil surface during the overpass [21]. The factors are mainly related to crop development (e.g., crop and residue cover), weather conditions (e.g., soil moisture content) and agricultural practices (e.g., soil crust and soil roughness).

When we only considered an NDVI threshold to mask vegetation, several single date S-2 images could detect the exposed soil for at least 40% of the cropland surface (Figure 8A). Yet, the SOC prediction for none of these single-date images reached an $R^2$ higher than 0.25, with RMSE that does not drop below 2.50 g C kg$^{-1}$ (Figure 4). If an NBR2$_{0.07}$ threshold was applied, the model performance increased, in some cases up to $R^2$ of 0.77 and RMSE dropped to 0.5 g C kg$^{-1}$, but the area that could be mapped dropped to less than 10% of the cropland (22 March 2019, Figures 4 and 9). Hence, the choice of acquisition date for achieving a good SOC prediction performance is crucial, while at the same time mosaicking can increase the area covered by the SOC models. Here, we, therefore, focused on an objective set of criteria for selecting the optimal acquisition dates to be included in an image composite for SOC prediction.

The soil remote sensing community widely uses NDVI thresholds to extract exposed soil [9,38,39]. However, Vaudour et al. [19] have demonstrated the inadequateness of the

NDVI index on its own for creating a temporal mosaic of exposed soil for the purpose of soil property prediction. This is in agreement with our findings. After all, the SOC prediction performance is poor for a composite based on NDVI only (composite B; Table 4, Figure 5). Moreover, SOC prediction performance is not even acceptable for composites using combined NDVI and NBR2 thresholds (composites C to E; Table 4, Figure 5). This is in contrast to Vaudour et al. [19] who have obtained acceptable SOC prediction in the Versailles Plain, France. This is likely caused by their prior removal of images acquired in winter, which was not done here. The sun zenith angle drops below 70° in winter scenes, which hampers the correct estimation of atmospheric parameters used for converting Level-1C to Level-2A product [40]. Consequently, the uncertainty of a Level-2A S-2 product is higher for these scenes [40]. In addition, winter acquisition dates witness high soil surface moisture, which strongly affects the overall shape of the reflectance spectra of a pixel. This effect is not always filtered by an NBR2 threshold if soils are moist and covered with residues, as NBR2 shows a mixed reaction to soils where both crop residues and high soil moisture are combined [24]. This is in agreement with Daughtry and Hunt [23] who stated that remotely sensed estimates of crop residue cover are erratic and unreliable without a robust correction for scene moisture content. Hence, it is difficult to apply a single NBR2 threshold to extract bare soils, and we opted for arbitrary classes of NBR2. So far no conventional threshold for NBR2 has been discussed in the literature: for example, Castaldi et al. [4] have, by trial and error, tested various NBR2 thresholds to exclude spectra affected by high soil moisture content or crop residues, while Vaudour et al. [19] have applied NBR2 thresholds corresponding to the 1st quantile, the median and the 3rd quantile of the NBR2 distribution.

We have applied the greening-up method to select the most suitable acquisition date for each pixel. This method allowed narrowing down the number of spectra used for the PLSR model, by pinning down the period during which soils are most likely to be exposed and smooth. The soil pixels selected based on the greening-up criterium are likely to be in seedbed condition, i.e., residues have been plowed in and the soil is smooth after harrowing. The greening-up approach applied on its own (composite F) did not result in a correct SOC prediction either: the prediction performance dropped compared to the composites relying on the NDVI and NBR2 indices in synergy or individually (Table 4, Figure 5). By applying a strict NBR2 threshold, however, the quality of prediction greatly improved (composite I, $R^2 = 0.54 \pm 0.12$, RPD = $1.68 \pm 0.45$ and RMSE = $2.09 \pm 0.39$ g C kg$^{-1}$, Table 4). This is due to the fact that during the greening-up period, soils are without residues. Under such conditions, the NBR2 index appears to be reliable for masking moist soils. Hence, the combination of greening-up and NBR2 extracts smooth, bare soils that are dry. Further research is needed to test the robustness of the NBR2 to mask moist pixels for these soils without residues.

The SOC pixels predicted using composite I (the greening-up method combined with a strict NBR2 index) covered more than 62% of the arable cropland for images acquired during the two years. This exceeds at least threefold the amount of exposed soil pixels of the single date S-2 images which allowed for a similar SOC prediction accuracy, i.e., $R^2 > 0.5$ and RMSE < $2.00$ C kg$^{-1}$ (21 January 2019, 22 March 2019, 1 June 2020, 31 July 2020, 5 August 2020 and 8 August 2020, Figure 4). The months of April, May, August and September accounted in our case for the biggest increase in exposed croplands. This is in agreement with the crop calendar of the Walloon region, i.e., potatoes, sugar beet and maize are sown in April/May, and winter cereals are sown in September.

Rogge et al. [15] developed an automated process to overcome the issue of limited soil exposure in satellite images, the Soil Composite Mapping Processor (SCMaP). The output generated by SCMaP is the average reflectance per pixel of each spectral band over a variable time period. Such averaging allows for the reduction of variability in the exposed soils, caused by factors such as crop residue cover, moisture and roughness [15]. They used five-year time periods to create exposed soil composites with sufficient soil cover on the one hand, and comparable data products from 1984 to 2014 on the other [15]. The

longer time period was necessary to account for the lower repetition rate of the Landsat sensors (16 days), which can now be reduced to two years for Sentinel-2 (5 days when both S-2A and S-2B are used). For the same area, van Wesemael et al. [41] have shown that the change in SOC contents of croplands was negligible at on average 0.27 g C kg$^{-1}$ over a ten-year period [41]. SCMaP is also a flexible tool to produce season-specific composites, and its product has been used locally for SOC prediction by Žížala et al. [42]. Vaudour et al. [21] suggested the inclusion of only specific periods (i.e., spring versus autumn), as they provided the best results in the Versailles Plain. Yet, at larger scales, the regional phenology varies and such selection of single suitable sensing period might constrain the results. We, on the other hand, proposed a method where a limited amount of scenes is selected based on the greening-up period, and only such scenes were included in the final composite. We applied a simple algorithm relying on binarized NDVI information, which diminished the computational time. However, many simplifications were made, and our approach does not reach the complexity of for example the TIMESAT algorithm, which defines key phenology dates and retrieves a set of phenology metrics [43]. Nevertheless, the greening-up method can probably be applied in most regions of the world. We believe that the key to success lies in the combination of the SCMaP multiple-year approach which stabilizes the signal, with the greening-up approach that narrows down the number of satellite images, independently of the region. Additionally, the SCMaP derives thresholds for spectral indices by using temporal spectral data and existing land cover data [15]. This allows for automatic threshold selection for various regions of the world, thus bypassing the need for manual selection of NDVI and NBR2 thresholds, as was done here.

## 5. Conclusions

Several authors have used composite images to increase cropland area for which SOC content can be predicted. The surface conditions have, however, hindered the accuracy of the SOC prediction models. Hence, spectral indices are being used by many authors to discriminate between soils in suitable and unsuitable conditions. However, the amount of available spectral indexes on the multispectral Sentinel-2 is limited, and the width of the spectral bands does not allow for a straightforward detection of disturbing effects such as crop residues, soil moisture and soil roughness. To select the most appropriate pixels to be included in a composite image for SOC prediction, we have explored the potential of pinning down the right acquisition date for each pixel based on the crop calendar. We defined as greening-up the instant for which the crop has been sown but has not yet emerged. This means that an eventual crust and crop residues have been plowed in and soils are in seedbed condition (i.e., smooth). This is the closest we are able to get to the pure soil signal of the surface spectrum. Once the crop residues were removed by selecting the greening-up moment, we applied the NBR2 index in order to remove pixels with high moisture contents. This greening-up-NBR2 combination applied as a threshold for a two-year series of S-2 images provided a SOC prediction model with a fairly good performance ($R^2$ = 0.54 $\pm$ 0.12, RPD = 1.68 $\pm$ 0.45 and RMSE = 2.09 $\pm$ 0.39 g C kg$^{-1}$), and covered 62% of the cropland. Overall, the greening-up-NBR2$_{0.07}$ synergy is a relatively simple (based on NDVI and NBR2), automated and objective method for accomplishing a trade-off between model performance and surface cover.

**Author Contributions:** Conceptualization, K.D. and B.v.W.; methodology, K.D. and B.v.W.; software, K.D.; validation, K.D. and B.v.W.; formal analysis, K.D.; investigation, K.D.; resources, B.v.W.; data curation, B.v.W. and K.D.; writing—original draft preparation, K.D.; writing—review and editing, K.D., U.H. and B.v.W.; visualization, K.D.; supervision, B.v.W.; project administration, B.v.W.; funding acquisition, B.v.W. and U.H. All authors have read and agreed to the published version of the manuscript.

**Funding:** Klara Dvorakova is a research fellow of the Fonds de la Recherche Scientifique—FNRS. The research is carried out in the framework of the Worldsoils project financed by the European Space Agency and the support is gratefully acknowledged.

**Acknowledgments:** We thank Marco Bravin of the Earth and Life Institute of the Université Catholique de Louvain (UCLouvain) for the essential organic carbon measurements.

**Conflicts of Interest:** The funders had no role in the design of the study; in the collection, analyses, or interpretation of data; in the writing of the manuscript, or in the decision to publish the results. The authors declare no conflict of interest.

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
