# Peer review of "Sentinel-2 Exposed Soil Composite for Soil Organic Carbon Prediction"

_remotesensing, doi:10.3390/rs13091791_

Round 1

Reviewer 1 Report

The article presents a simple but efficient workflow for bare soil composite creation and application of the composite to SOC prediction. The article is novel and well-prepared presenting a hot topic. I have just two two questions/suggestions.

Line 172-177. Did you consider some automatic methods for finding the right index thresholds? E. g. Otsu´s method, histogram analysis. Can be searching for a threshold somehow automatized to support the region independency (lines 493-494)? I suggest adding a paragraph concerning automatic searching for a threshold the at the end of the discussion.

Could you add a paragraph commenting the mentioned uncertainty (lines 180-182) of NDVI and NBR2 thresholding and a comparison of threshold values with previous articles to discussion? I think that is a key and most tricky step for a bare soil-composite creation.

Reviewer 2 Report

This manuscript entitled “Sentinel-2 exposed soil composite for soil organic carbon prediction” proposes a method for deriving a bare soil composite based on the ‘greening-up’ period combined with NBR2.

Overall this is a well-written article, which confirms previous recent results obtained from temporal composites in the literature, and proposes an alternate approach relying on the NDVI temporal profiles.

The idea consists in avoiding relying on external data such as S1-derived soil moisture maps as previously used by [19] to rank dates to be included in a temporal mosaic.

The main concern for this manuscript is the absence of observed soil surface characteristics and crops: it solely relies on assumptions about the presence of crop residues, soil moisture and soil roughness. This should be emphasized, as well as the possible limitations of the “greening-up” approach: for instance, the scarce availability of images according to weather conditions (minimum is inferred from two images only in 2019). At least,  weather conditions should be discussed.

The sentence in conclusion stating that “the amount of spectral indexes on the multispectral Sentinel-2 is limited, and the width of the spectral bands does not allow for a straightforward detection of disturbing effects such as crop residues, soil moisture and soil roughness” is not supported by the observations. Moreover, the size of the sample for which best results are obtained is rather limited, and these results should be considered with some caution. Conclusion should be rephrased.

In conclusion, I recommend minor revision.

In detail:

Lines 74-79 and line 83: this is not exactly a soil moisture spectral index but, according to the approaches tested (pixelwise or per date), either the volumetric soil moisture per pixel or the average per-date volumetric soil moisture; also amongst the approaches tested by [19], the best areal compromise was considered.

Line 79-80: over 2 years for an area characterized by 4-years crop rotations

Line 88-90: such argument is not obvious, as nothing insures that the acquired images during these ‘optimal’ windows are performing for SOC prediction.

Lines 83-85: Soil moisture maps are made available on the theia website, no prior information is required.

Lines 134-135: are there calcaric soils amongst the 137 samples?

Line 152 and Fig.1 : the image of 8 August 2020 has clouds in its southeastern part. “Cloud free” but not at the scale of the tile?

Line 181-182: how did you select these ‘arbitrary’ thresholds?

Lines 268-275: hence the areal percentages of exposed soil are derived from these 18000 points (~0.1% of the cropland area)?

Table 3: the size of the I sample is small; did you examine its spatial spread across the area?

Figure 4: please also add the RPD

Although soil moisture may not been available, are there rainfall events shortly before each of these dates? Particularly those for which the threshold of 0.07 was too restrictive?

Line 439: these authors actually tested a number of mosaics made of several dates including Autumn and Winter dates, but the best performing mosaics were always composed of Spring dates

Lines 468-472: in my opinion, the percentage of bare soil that can be retrieved is not readily comparable from an agricultural cropland region to another, even more while crop rotations last 4 years (in [19]) and not 3 years as in the Walloon region. In addition the areal percentage is not calculated on the same basis (for [19], from all the pixels included in the mosaic, which was effectively built). Was the exhaustive areal percentage computed from all bare soil pixels included in a given mosaic (one of the 9 tested) compared with that based on the 18000-pixels network?

Line 478: readers might be confused and interpret that Rogge et al tested their approach for SOC content prediction; this should be specified. Zizala et al 2019 tested Rogge’s compositing approach for the purpose of SOC content prediction, but at local scale.

Lines 481-482: does it exceed the time when SOC content can be considered “stable” in the Walloon region?

Line 490-491: algorithm based on binarized information also performed by [19]
